# Monitoring of cardiovascular physiology augmented by a patient-specific biomechanical model during general anesthesia. A proof of concept study

Arthur Le Gall[1,2,3,4], Fabrice Vallée[1,2,3,4], Kuberan Pushparajah[5], Tarique Hussain[6], Alexandre Mebazaa[3,4], Dominique Chapelle[1,2], Étienne Gayat[3,4], Radomír Chabiniok[1,2,5,7] *

**1** Inria, Paris, France, **2** LMS, École Polytechnique, CNRS, Institut Polytechnique de Paris, Paris, France, **3** Anesthesiology and Intensive Care Department, Lariboisière - Saint Louis - Fernand Widal University Hospitals, Paris, France, **4** INSERM, Paris, France, **5** School of Biomedical Engineering & Imaging Sciences, St Thomas' Hospital, King's College London, London, United Kingdom, **6** Department of Pediatrics, Division of Pediatric Cardiology, UT Southwestern Medical Center, Dallas, TX, United States of America, **7** Department of Mathematics, Faculty of Nuclear Sciences and Physical Engineering, Czech Technical University in Prague, Prague, Czech Republic

* radomir.chabiniok@inria.fr

**Data Availability Statement:** All relevant data are within the manuscript and its Supporting Information files.

## Abstract

During general anesthesia (GA), direct analysis of arterial pressure or aortic flow waveforms may be inconclusive in complex situations. Patient-specific biomechanical models, based on data obtained during GA and capable to perform fast simulations of cardiac cycles, have the potential to augment hemodynamic monitoring. Such models allow to simulate Pressure-Volume (PV) loops and estimate functional indicators of cardiovascular (CV) system, e.g. ventricular-arterial coupling ($V_{va}$), cardiac efficiency (CE) or myocardial contractility, evolving throughout GA. In this prospective observational study, we created patient-specific biomechanical models of heart and vasculature of a reduced geometric complexity for n = 45 patients undergoing GA, while using transthoracic echocardiography and aortic pressure and flow signals acquired in the beginning of GA (baseline condition). If intraoperative hypotension (IOH) appeared, diluted norepinephrine (NOR) was administered and the model readjusted according to the measured aortic pressure and flow signals. Such patients were a posteriori assigned into a so-called hypotensive group. The accuracy of simulated mean aortic pressure (MAP) and stroke volume (SV) at baseline were in accordance with the guidelines for the validation of new devices or reference measurement methods in all patients. After NOR administration in the hypotensive group, the percentage of concordance with 10% exclusion zone between measurement and simulation was >95% for both MAP and SV. The modeling results showed a decreased $V_{va}$ (0.64±0.37 vs 0.88±0.43; p = 0.039) and an increased CE (0.8±0.1 vs 0.73±0.11; p = 0.042) in hypotensive vs normotensive patients. Furthermore, $V_{va}$ increased by 92±101%, CE decreased by 13±11% ($p < 0.001$ for both) and contractility increased by 14±11% ($p = 0.002$) in the hypotensive group post-NOR administration. In this work we demonstrated the application of fast-running patient-specific biophysical models to estimate PV loops and functional indicators of CV system using

**Funding:** A.L.G. and F.V. were employed full-time by a research program "Poste d'accueil APHP" funded 50% by Assistance Publique - Hopitaux de Paris and 50% by Inria. R.C. and K.P. acknowledge the support of Wellcome/EPSRC Centre for Medical Engineering [WT 203148/Z/16/Z]. R.C., D.C. and T. H. acknowledge the support of Inria-UTSW Associated Team TOFMOD. R.C. additionally acknowledges support of the Ministry of Health of the Czech Republic (project No. NV19-08-00071).

**Competing interests:** I have read the journal's policy and the authors of this manuscript have the following competing interests: A.L.G., F.V., D.C. and R.C. are co-owners of the patent entitled "Dispositif cardiaque" (number 1758006, 2017). A research license agreement is currently ongoing between the Anesthesiology and intensive care department of Lariboisière hospital, Paris, France and Deltex Medical, Chichester, UK. This does not alter our adherence to PLOS ONE policies on sharing data and materials.

clinical data available during GA. The work paves the way for model-augmented hemodynamic monitoring at operating theatres or intensive care units to enhance the information on patient-specific physiology.

## Introduction

Cardiac physiology is a delicate balance between extrinsic (e.g. preload or afterload) and intrinsic (e.g. contractility or electrical activation) properties of the heart. Cardiovascular (CV) failure is the third reason for entering the intensive care unit (ICU) and the second cause of in-ICU death [1]. Furthermore, it is estimated that around 230 million major surgical procedures under general anesthesia (GA) are performed each year worldwide [2] and perioperative CV events remain the main cause of postoperative death [3]. CV management during GA or at critical care includes CV monitoring based on (and not limited to) arterial pressure and cardiac output (CO) measurements [4, 5].

The simultaneous evaluation of the ventricular pressure and volume (PV loop) allows a functional interpretation of pathophysiological conditions, such as quantifying myocardial energetic expenditure or ventricular-arterial coupling [6] (see Fig 1). In some complex cases, the PV loop analysis can bring additional insight in to the current physiological state [7]. However, as it requires invasive intraventricular pressure measurement, its usage is not convenient during monitoring.

Patient-specific CV modeling provides a numerical representation of the CV system in individual patients—a "numerical avatar"—which is becoming a powerful diagnostic or therapeutic tool. For example, it allows to access the PV loop [8–10], predict the success of cardiac resynchronization therapy [11, 12], or estimate myocardial stiffness and contractility or vascular resistance [13–15] including under various physiological conditions [16, 17], see also reviews [18, 19] and references therein. However, a requirement of fast analysis alongside with restricted data availability make its implementation within CV monitoring rather challenging.

In the present study, we aimed to evaluate the feasibility of using a monitoring framework augmented by a biophysical model to obtain and interpret the simulated PV loops and some CV functional quantities, while using only data readily-available during neuroradiological procedure.

## Methods

This prospective and non-interventional cohort study was held in a university hospital in Paris, and followed the STROBE guidelines for conducting observational studies.

### Patients monitoring and data collection

Patients scheduled for an intracranial endovascular procedure were selected for inclusion in this study. Only the patients for whom continuous arterial pressure and CO monitoring was indicated for clinical purposes were included. This study was approved by the appropriate Institutional Review Board—ethical committee of the Société de Réanimation de Langue Française (CE-SRLF 14-34)—which waived the need for written informed consent. Consequently, oral informed consent was obtained from all subjects after providing a protocol information letter. Every subject had the possibility to withdraw from the study at any time.

During neuroradiological procedure, GA was induced and maintained by total intravenous anesthesia using propofol (75-150 mg/kg/min) and remifentanil (0.2-0.5 $\mu$g/kg/min). Oro-

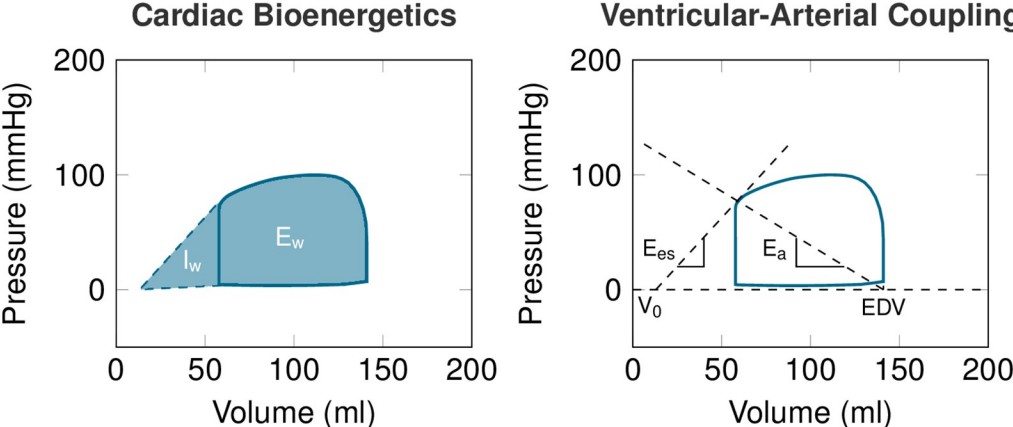

**Fig 1. Example of PV loop and its interpretation.** Cardiac bioenergetics (left): internal work ($I_w$) associated with the potential energy or the energetic expenditure necessary to reach optimal conditions for ejection; external work ($E_w$) associated with the energetic expenditure of the ejection; cardiac efficiency (CE) defined as the ratio $CE = \frac{E_w}{E_w + I_w}$. Ventricular-arterial coupling (right): ventricular elastance $E_{es}$ (slope of the end-systolic pressure-volume relationship, ESPVR, at end-systolic pressure-volume point); arterial elastance $E_a = \frac{ESP}{SV}$ with ESP being end-systolic pressure and SV = EDV − ESV the stroke volume (subtraction of end-diastolic and end-systolic volumes).

tracheal intubation was facilitated using 0.5 mg/kg atracurium and followed, if needed, by continuous infusion of 0.5 mg/kg/h atracurium. After intubation, ventilation was established to reach an end-tidal $CO_2$ concentration of 35-38 cm $H_2O$ using a tidal volume of 6-8 ml/kg of body weight.

After GA induction, the monitoring devices were installed. Transesophageal Doppler probe (TED) was inserted into the esophagus and connected to the CombiQ monitor (Deltex medical, Chichester, UK). A transthoracic echocardiography (TTE) was performed at the beginning of the interventional procedure. A radiopaque wire was advanced from the femoral artery through the aorta up to cerebral arteries. Invasive arterial pressure was recorded by connecting a fluid-filled mechanotransducer (TruWave, Edwards Lifescience, Irvine, CA, USA), as previously described in [20]. For research purposes, data were recorded when the pressure catheter was in the ascending aorta.

Our standard procedure for management of intraoperative arterial hypotension (IOH), defined as the fall of mean arterial blood pressure by 20% as compared to the awake value, includes: 1) titration of saline solution by 250 ml step s to optimize CO; and 2) in case of persistent IOH despite the fluid expansion, titration of diluted norepinephrine (NOR) (5$\mu$g/ml) by 5 $\mu$g steps to restore the blood pressure. This management was not modified for this research project. We separated a posteriori the population into so-called hypotensive and normotensive groups according to whether they received NOR.

**Validation group.** In order to compare the PV loops obtained by the model to invasively acquired PV loops, we analyzed four additional patients with pressure and volume measured in their systemic ventricles by pressure catheter and magnetic resonance imaging (MRI), respectively. Validation subjects #1-2 had a single-ventricular physiology (Fontan circulation) and underwent combined cardiovascular MRI and heart catheterization procedure for progressive symptoms of exercise intolerance. Validation subjects #3-4 were patients with repaired tetralogy of Fallot who underwent cardiovascular MRI and catheterization before pulmonary valve replacement.

All XMR exams were performed under GA. During the catheterization, pressure signals (aortic, ventricular, venae cavae, and pulmonary capillary wedge pressures) were obtained.

Simultaneously, cardiac volumes and 2D flows (ascending and descending aorta) were acquired using cardiovascular MRI (CMR). The obtained CMR data were post-processed into 0D signals of time-vs-flow or time-vs-ventricular volumes.

The data collections in the patients with Fontan circulation were performed at Evelina London Children's Hospital, King's College London, under the ethical approval of institutional ethics committee, London UK (Ethics Number 09H0804062). The data collection in tetralogy of Fallot patients were performed at Children's hospital, UT Southwestern Medical Center Dallas, under the ethical approval UT Southwestern IRB (STU 032016-009).

## Biomechanical model of cardiovascular system for monitoring purposes

The biomechanical model of heart and vasculature used in this study is described in [21]. It is a combination of a biomechanical heart [22, 23] and a Windkessel circulation model [24].

The passive component of the myocardium is inspired by [25] with a hyperelastic potential in the form

$$W_e = C_0 e^{C_1(J_1-3)^2} + C_2 e^{C_3(J_4-1)^2}, \tag{1}$$

with $J_1$ and $J_4$ being reduced invariants of the left Cauchy-Green tensor $C$, given by $J_1 = \mathrm{trace}(C)(\det(C))^{-\frac{1}{3}}$ and $J_4 = fib \cdot C \cdot fib (\det(C))^{-\frac{1}{3}}$ (with $fib$ being the unit vector in the myocardial fiber direction). We used the parameters $C_0$ = 665 Pa, $C_1$ = 2.4 Pa, $C_2$ = 103 Pa and $C_3$ = 5.5 Pa, which allow to fit the experimentally measured end-diastolic pressure volume relationship (EDPVR) [26] in a reference healthy human. The passive potential (Eq (1)) is then multiplied by a "stiffness multiplication factor"—the only parameter used in adjusting the passive part of the model to a given patient (passive tissue stiffness). The active component of the model—representing actin-myosin interaction and formation of cross-bridges—is based on Huxley's sliding filament theory [27, 28], albeit with an extension allowing to represent the Frank-Starling mechanism [23]. The active stress $\tau_c$ and active stiffness $k_c$ in sarcomeres with the extension $e_{fib} = \frac{L}{L_0} - 1$ (where $L$ and $L_0$ represent the actual and reference sarcomere lengths, respectively) generated in the sarcomere are given by

$$\begin{cases} \dot{k}_c = -(|u| + \alpha|\dot{e}_{\mathrm{fib}}|)k_c + n_0(e_{\mathrm{fib}})k_0|u|_+ \\ \dot{\tau}_c = -(|u| + \alpha|\dot{e}_{\mathrm{fib}}|)k_c + n_0(e_{\mathrm{fib}})\sigma_0|u|_+ + k_c\dot{e}_{\mathrm{fib}}. \end{cases} \tag{2}$$

The asymptotic active stress $\sigma_0$ and stiffness $k_0$, generated by the sarcomere, are directly related to myocardial contractility, while taking into account the effect of actin-myosin overlap using a Frank-Starling law function $n_0(e_{fib})$ with value s between 0 and 1 (the maximum value for the optimal fiber extension and optimal overlap of actin and myosin chains), for details see [23]. The activation of the sarcomeres is modeled using an activation function $u$, which is positive when the tissue is electrically activated with the maximum value of $35s^{-1}$ (given by the rate of active stress generation [29]), $|u|_+$ being defined as $\max(u, 0)$, and $|u|_-$ as $\max(-u, 0)$. The parameter $\alpha$ governs the cross-bridge destruction rate due to rapid length changes.

The LV geometry was reduced to a thin-walled sphere as described in [21]. While the geometry and kinematics are simplified, all physical and physiological components are preserved. The model is then solved for the unknown displacement in the radial direction $y(t) = R(t) - R_0$, where $R(t)$ and $R_0$ stand for the actual (at time $t$) and reference (stress-free) radii of the sphere (from the center to the mid-wall), and unknown intra-ventricular pressure $P$ (and active stiffness and stress as internal variables). Using the fiber extension $e_{fib} = \frac{y}{R_0}$, the kinematics of the model can be rewritten as: $R = R_0(1 + e_{fib})$. Likewise, due to tissue incompressibility

the thickness of the myocardium $d = d_0(1+ e_{fib})^{-2}$, with $d_0$ being the wall thickness in the reference configuration. Ventricular volume is then given by $V = \frac{4}{3}\pi(R - d/2)^3$.

The circulation is represented by a Windkessel model containing proximal and distal capacitances ($C_p$ and $C_d$) and resistances ($R_p$ and $R_d$) connected in series, with the distal part representing the majority of the vascular resistance—typically ten times higher than in the proximal part. The Windkessel model equations read:

$$\begin{cases} C_p\dot{P}_{ao} + (P_{ao} - P_d)/R_p & = Q_{ao} \\ C_d\dot{P}_d + (P_d - P_{ar})/R_p & = (P_{ve} - P_d)/R_d, \end{cases} \tag{3}$$

with $Q_{ao}$ being the flow through aortic valve and $P_{ao}, P_d, P_{ve}$ representing aortic, distal arterial and venous pressures, respectively.

We remark that even though the geometry of the model used in this work is reduced to a sphere, the adjustment of the cavity size, myocardial mass and biophysical properties of the tissue allows to tailor the model to individual patients. Thanks to such a reduction of geometric complexity, the proposed formulation allows to use patient-specific cardiovascular modeling in close to real-time setting—a single heart beat being simulated within a few seconds—with standard computational resources.

## Calibration of the model to data of individual patients

The generic model was turned into patient- and physiology-specific regime by a calibration procedure, during which the model parameters were manually adjusted according to the measured clinical data (see Table 1). The sequential calibration procedure consists of:

1. Adjustment of the parameters of Windkessel circulation model after imposing the flow in ascending aorta, with the objective of matching the measured aortic pressure with the simulation. As the monitoring data contain only velocity in the descending aorta, this waveform was scaled by using stroke volume (SV) obtained by TTE in the beginning of the procedure, in order to obtain a surrogate for the ascending aortic flow signal.

2. Adjustment of the left ventricular (LV) geometry (LV volume and myocardial mass) according to the TTE measurements taken at end-diastole. The assumed LV volume at zero pressure level (the so-called reference configuration), was given according to [26] by EDV · (0.6-0.006 · EDP), where EDP was the assumed end-diastolic ventricular pressure, see next point (pressure and volume in the formula are given in mmHg and ml, respectively).

3. Adjustment of the passive tissue stiffness of myocardium aiming at obtaining the EDV as measured by TTE while applying the ventricular EDP. Not having an access to the ventricular or atrial pressure, we used a semi-quantitative method to classify LV filling pressure as either high or normal [30], and we arbitrarily prescribed EDP value of 15 or 7 mmHg, respectively.

4. Adjustment of timing of the electrical activation by using the measured ECG.

5. Adjustment of the myocardial contractility in the model to reach the stroke volume (SV) as in the data.

The model calibration was performed for all patients at baseline. If NOR was requested during GA, we re-adjusted only the parameters that are expected to be involved by NOR (i.e. Windkessel model, timing of heart activation and myocardial contractility, see Table 2).

**Table 1. Imaging data procedure: Transthoracic echocardiography mesurements.**

| Parasternal long axis | Left Ventricular Posterior Diameter | (LPWD) |
|---|---|---|
| | Aortic Root Diameter | (ARD) |
| | Septum Diameter | (SD) |
| | Left Ventricular End-Diastolic Diameter | (LVEDD) |
| Apical 4-chamber | E wave | |
| | A wave | |
| | E' wave (Tissue Doppler Imaging) | |
| | Left Ventricular End-Diastolic Surface | (LVEDS) |
| | Left Ventricular End-Systolic Surface | (LVESS) |
| | Left Atrial Surface | (LAS) |
| Apical 5-chamber | Velocity Time integral of Left Ventricular Outflow Tract | (VTI) |

## Objectives

The first objective was to demonstrate that the calibrated models at baseline accurately represent the patients' data by conducting an equivalence study. The second objective was to employ the augmented hemodynamic monitoring to quantify the alterations of cardiovascular state during IOH and after administering NOR to restore blood pressure.

## Judgement criteria

The primary endpoint was to test the equivalence between the simulated and measured aortic pressures and flow for the population. The mean, systolic and diastolic aortic pressures (MAP, SAP, DAP), and SV were used. The secondary endpoints were to compare the hypotensive and normotensive patients, and the hypotensive patients during the restoration of blood pressure by NOR in the following sense: 1) Distal resistance ($R_d$) and capacitance ($C_d$) of the Windkessel model with the calculated systemic vascular resistance (SVR) and total arterial compliance ($C_{tot}$), respectively; 2) myocardial contractility; 3) simulated indicators of ventricular-arterial coupling ($V_{va}$); and 4) simulated indicators of heart bioenergetics (Table 3).

## Statistics

We designed an equivalence study to validate the ability of our framework to reproduce the aortic pressure and CO. We followed the extended CONSORT guidelines for reporting equivalence and non-inferiority studies [31]:

**Table 2. List of parameters used for calibration.** In bold are the parameters which were re-calibrated after norepinephrine administration according to the new physiology state.

| | | |
|---|---|---|
| 1 | Sphere radius at reference configuration | |
| 2 | Sphere thickness at reference configuration | |
| 3 | Atrial pressure | |
| **4** | **Heartbeat duration** | |
| **5** | **Time of ventricular activation** | |
| **6** | **Duration of electrical activation** | |
| 7 | Myocardial stiffness factor | |
| **8** | **Proximal Windkessel resistance** | |
| **9** | **Distal Windkessel resistance** | |
| **10** | **Distal Windkessel capacitance** | |
| **11** | **Myocardial contractility** | |

**Table 3. Calculations of heart function indicators.** MAP, mean aortic pressure; SAP, systolic aortic pressure; DAP, diastolic aortic pressure; LVEDS, left ventricular end-diastolic surface; LVESS, left ventricular end-systolic surface; LVESP, left ventricular end-systolic pressure; AOD, left ventricular outflow tract diameter; HR, heart rate; $V_0$, intersection of end-systolic pressure volume relationship with volume axis.

| | |
|---|---|
| Stroke Volume | $SV = \pi \frac{AOD^2}{4} \cdot VTI$ |
| Left Ventricular End-Diastolic Volume | $LVEDV = \sum_{i=1}^{n} \text{pixel spacing} \cdot \pi \frac{LVEDS^2}{4}$ |
| Left Ventricular End-Systolic Volume | $LVESV = \sum_{i=1}^{n} \text{pixel spacing} \cdot \pi \frac{LVESS^2}{4}$ |
| Left Ventricular Ejection Fraction | $LVEF = \frac{LVEDV - LVESV}{LVEDV}$ |
| Pulse Pressure | $PP = SAP - DAP$ |
| Cardiac Output | $CO = SV \cdot HR$ |
| Total Arterial Compliance | $C_{tot} = \frac{SV}{PP}$ |
| Systemic Vascular Resistance | $SVR = \frac{MAP}{CO}$ |
| Arterial Elastance | $E_a = \frac{LVESP}{SV}$ |
| Ventricular Elastance | $E_{es} = \frac{LVESP_{highPreload} - LVESP_{lowPreload}}{LVESV_{highPreload} - LVESV_{lowPreload}}$ |
| Ventricular-Arterial Coupling | $V_{va} = \frac{E_a}{E_{es}}$ |
| Internal Work | $I_w = \frac{LVESP(LVEDV - V_0)}{2}$ |
| External Work | $E_w = \text{area under PV loop}$ |
| Cardiac Efficiency | $CE = \frac{E_w}{E_w + I_w}$ |

1. Rationale and choice of equivalence margins: according to the guidelines for the validation of a new arterial pressure device [32], we set the equivalence margins for the simulated aortic pressure to 13 mmHg. The equivalence margins for the simulated CO, using the percentage error $PE = 100 \cdot 1.96 \cdot \frac{SD_{sim-meas}}{Mean_{meas,sim}}$, was set to 30%, a margin consistent with the ability of current physiological monitors to measure the trends in CO (as reviewed in [33]). The coefficient error ($CErr = \frac{SD}{Mean} \cdot \frac{1}{\sqrt{nob}}$, $nob$ being the number of heart beats considered for calculation) was also calculated.

2. Sample size calculation: With a first-order error $\alpha = 0.025$ and a power $(1 - \beta) = 0.99$, the number of patients requested to include in the equivalence study was 45.

3. Confidence interval analysis: We provided a Bland-Altman plot for repeated measurements to represent the bias with the reference method. We tested the equivalence between the simulation and the measurement using the Two One-Sided Test (TOST, [34]). This test postulates that accepting the $H_0$ hypothesis implies that there exists a difference between the two tested means, and accepting the $H_1$ hypothesis ($p < 0.05$) implies that the two tested means are equivalent.

We compared the characteristics of the normotensive group with the characteristics of the hypotensive group at baseline by using the $\chi^2$-test for categorical variables and by the Wilcoxon test for continuous variables. In the hypotensive group, we further analyzed the variation of the parameters of the models and the results of the simulations from baseline to the maximum effect of NOR using the Wilcoxon test. The continuous variables were presented as mean ± standard deviation and the categorical variables as count (%).

The simulations were performed using an in-house implementation of the model in MATLAB (The MathWorks Inc, Natick, Massachusetts). The statistical analysis was performed using R (The R Foundation for Statistical Computing, Vienna, Austria).

**Table 4. Population characteristics and comparison between the normotensive and hypotensive group.** Results are expressed as mean ± standard deviation or count (percentage).

| Patient's data | | | All | Normotensive | Hypotensive | P-val |
|---|---|---|---|---|---|---|
| | | | n = 45 | n = 29 | n = 16 | |
| **Demographic** | | | | | | |
| Age | | (years) | 51±13 | 53±14 | 49±12 | 0.315 |
| Sex F | | n (%) | 21(46) | 11(39) | 10(63) | 0.997 |
| Weight | | (kg) | 73±15 | 76±17 | 69±11 | 0.292 |
| Height | | (cm) | 168±9 | 169±10 | 166±8 | 0.329 |
| **Comorbidities** | | | | | | |
| Hypertension | | n (%) | 14(31) | 10(39) | 4(25) | 0.957 |
| Diabetes | | n (%) | 2(4) | 1(4) | 1(6) | 0.516 |
| Dyslipidemia | | n (%) | 5(11) | 4(14) | 1(6) | 0.662 |
| Myocardial infarction | | n (%) | 1(2) | 1(4) | 0(0) | 0.468 |
| **Transthoracic echocardiography** | | | | | | |
| Ejection Fraction | | (%) | 59±9 | 57±9 | 61±8 | 0.127 |
| Wall Thickness | | (cm) | 0.78±0.13 | 0.83±0.13 | 0.76±0.14 | 0.69 |
| Aortic root diameter | | (cm) | 1.91±0.19 | 1.94±0.21 | 1.88±0.17 | 0.242 |
| End-Diastolic Volume | | (ml) | 129±25 | 132±25 | 126±25 | 0.681 |
| Left Atrial Volume | | (ml) | 54±5.4 | 56±3.8 | 52.4±7 | 0.682 |
| E-wave | | $(cm.s^{-1})$ | 76±24 | 70±21 | 82±26 | 0.105 |
| A-wave | | $(cm.s^{-1})$ | 61±18 | 64±18 | 59±15 | 0.432 |
| E'-wave (TDI) | | $(cm.s^{-1})$ | 12±5 | 11±3 | 13±5 | 0.352 |
| Velocity-Time Integral | | (cm) | 23±5 | 23±5 | 23±5 | 0.86 |
| Stroke Volume | | (ml) | 76±17 | 75±15 | 77±19 | 0.681 |
| **Hemodynamic** | | | | | | |
| Mean Pressure | | (mmHg) | 81±13 | 85±14 | 75±10 | <0.001 |
| Systolic Pressure | | (mmHg) | 114±19 | 118±20 | 109±16 | <0.001 |
| Diatolic Pressure | | (mmHg) | 60±10 | 62±11 | 56±7 | <0.001 |
| Stroke Volume (TED) | | (ml) | 73±18 | 70±15 | 76±22 | <0.001 |
| **Biology** | | | | | | |
| Potassium | | $mmol.l^{-1}$ | 4.1±0.4 | 4.1±0.5 | 4.1±0.3 | 0.962 |
| Serum Creatinine | | $\mu mol.l^{-1}$ | 70±16 | 73±18 | 65±12 | 0.242 |
| **Treatment** | | | | | | |
| Total Fluid Infusion | | ml | 1027±464 | 1079±417 | 950±612 | 0.475 |

## Results

Between November 1, 2016 and October 30, 2017, 45 patients were included (Table 4). Among them, 16 patients (36%) received at least one NOR administration to treat IOH and were included in the hypotensive group. The remaining 29 patients (64%) remained hemodynamically stable during the data recording and were included in the normotensive group. Main parameters of the models after calibrations to patients' data and the interpretation of PV loops are presented in Table 5. Fig 2 shows examples of the models confronted to the data. Fig 3 displays an example of simulated PV loop.

### Equivalence between clinical data and the calibrated models

The Bland-Altman plots for repeated measurements in Fig 4A and 4C demonstrate that the simulated and the measured aortic pressure and flow at baseline were concordant. The

**Table 5. Results of model calibration procedure for the entire population, and comparison between the normotensive and hypotensive group.** Results are expressed as mean ± standard deviation.

| Model parameters and simulation results | | All | Normotensive | Hypotensive | P-val |
|---|---|---|---|---|---|
| | | n = 45 | n = 29 | n = 16 | |
| **Model parameters** | | | | | |
| Ventricular volume | (ml) | 69±14 | 70±14 | 66±13 | 0.581 |
| at reference configuration | | | | | |
| Wall thickness at reference configuration | (cm) | 0.96±0.17 | 0.93±0.15 | 1±0.19 | 0.404 |
| at reference configuration | | | | | |
| Radius of ventricle from center | | | | | |
| to mid-wall at reference configuration | (cm) | 3±0.2 | 3.01±0.19 | 3±0.21 | 0.842 |
| Heartbeat duration | (ms) | 9501±181 | 9621±182 | 9291±182 | 0.433 |
| Distal Resistance | $(10^8\,Pa \cdot s \cdot m^{-3})$ | 1.25±0.33 | 1.33±0.35 | 1.1±0.25 | 0.014 |
| Distal Capacitance | $(10^{-8}\,m^{3.}\,Pa^{-1})$ | 1.25±0.38 | 1.2±0.4 | 1.35±0.31 | 0.09. |
| Contractility | (kPa) | 91±23 | 94±27 | 84±14 | 0.302 |
| **Ventricular-arterial coupling** | | | | | |
| Arterial elastance ($E_a$) | $(10^8\,Pa \cdot m^{-3})$ | 1.76±0.45 | 1.86±0.42 | 1.59±0.45 | 0.095 |
| Ventricular elastance ($E_{es}$) | $(10^8\,Pa \cdot m^{-3})$ | 2.76±1.52 | 2.52±1.15 | 3.2±1.99 | 0.182 |
| Ventricular-arterial coupling ($V_{va}$) | (unitless) | 0.8±0.4 | 0.88±0.43 | 0.64±0.37 | 0.039 |
| **Cardiac bioenergetics** | | | | | |
| External work ($E_w$) | (Joules) | 0.96±0.3 | 0.99±0.3 | 0.91±0.31 | 0.365 |
| Internal work ($I_w$) | (Joules) | 0.32±0.19 | 0.38±0.21 | 0.23±0.12 | 0.009 |
| Cardiac efficiency (CE) | (unitless) | 0.75±0.11 | 0.73±0.11 | 0.8±0.1 | 0.042 |

simulated MAP, SAP, DAP, and SV were statistically equivalent to the measurements: −0.9 (95%-confidence interval $CI_{95=-1.7}$ to −0.1) mmHg for MAP; −1.2 ($CI_{95=-2.2}$ to −0.2) mmHg for SAP; 2.4 ($CI_{95=1.6}$ to 3.2) mmHg for DAP; and 0.1 ($CI_{95=-0.9}$ to 1.2)% of measured SV for SV ($p < 0.001$ for equivalence for all). Furthermore, the upper and the lower bounds of the confidence interval for the differences between measurements and simulations were within

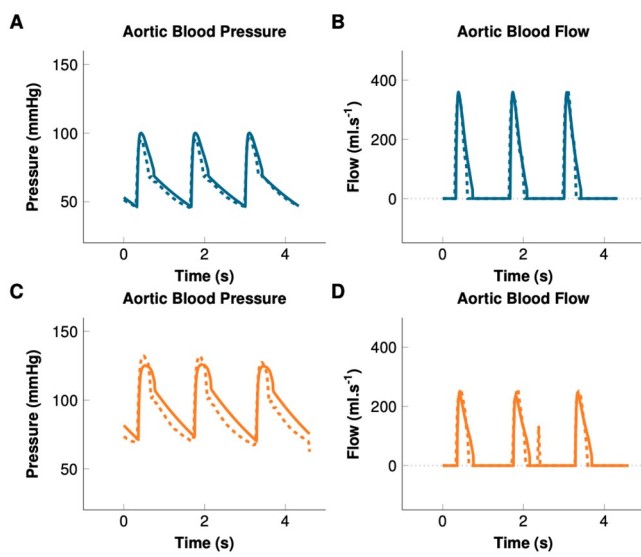

**Fig 2. Example of model calibration.** Solid lines represent the result of the patient-specific simulation. Dashed lines represent measured data. Blue: Hypotensive. Orange: Maximum effect of norepinephrine.

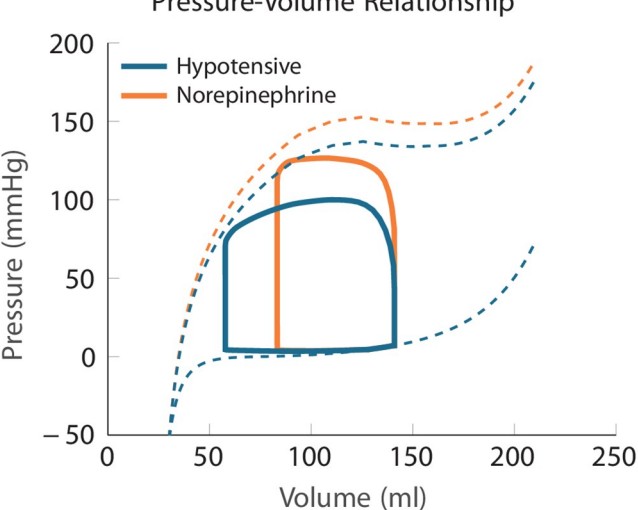

**Fig 3. Example of output of a patient-specific simulation for a hypotensive patient and at the maximum effect of norepinephrine.** Solid lines represent the dynamic pressure-volume relationship during a cardiac cycle—namely the Pressure-Volume (PV) loop. Dashed lines represent the static pressure-volume relationships—namely the End-Diastolic Pressure-Volume Relationship (EDPVR) and the End-Systolic Pressure-Volume Relationship (ESPVR). The EDPVR characterizes the ventricular volume for a given pressure at end-diastole. The ESPVR represents the ventricular pressure and volume at end-systole, prior to isovolumic relaxation. Note that the dynamic PV loop does not necessarily reach the theoretical static ESPVR curve, typically when cardiac cycle is too short.

the predefined margins of equivalence (Fig 5). The percentage error s for MAP, SAP, DAP, and SV were 6, 5, 8, and 18%, respectively. The coefficient error s for the simulation s were 0.61, 0.29, 1.27, and 0.5%, for MAP, SAP, DAP and SV, respectively. Finally, Fig 6 shows a statistically significant correlation between the measured and simulated indicators of arterial resistance and compliance.

## PV loop interpretation in normotensive vs hypotensive group before norepinephrine administration

Characteristics of normotensive patients were not different from hypotensive, except for the hemodynamic conditions before the administration of NOR (see Table 4). Specifically, the hypotensive patients had a lower blood pressure and a higher SV. The analysis of the values of the model parameters was consistent with these observations (see Table 5 and Fig 7), as the distal resistance was lower in the hypotensive group ($110\pm25$ vs $133\pm35$ MPa $\cdot$ s $\cdot$ m$^{-3}$; $p = 0.014$). The PV loop analysis showed that the ventricular-arterial coupling ($V_{va}$) was lower for the hypotensive than for normotensive group ($0.64\pm0.37$ vs $0.88\pm0.43$; $p = 0.039$). The internal work ($I_w$) was lower and the cardiac efficiency (CE) was higher in the hypotensive group ($0.23\pm0.12$ vs $0.38\pm0.21$ Joules; $p = 0.009$ and $0.8\pm0.1$ vs $0.73\pm0.11$; $p = 0.042$, for $I_w$ and CE, respectively), see Table 5 and Fig 7.

## Interpretation of the norepinephrine effects in the hypotensive group

The effect of NOR was confirmed by the changes in measured pressures and flow (MAP, SAP and DAP increased by $30\pm15$, $23\pm12$ and $27\pm13$%, respectively, whereas SV decreased by $14\pm9$%; $p < 0.001$ for all). Fig 2 shows an example of the adjusted calibration after NOR administration in a hypotensive patient. The adequacy between the simulations and measurements was confirmed by the 4-quadrant plots (Fig 4B and 4D) (>95% concordance with 10%

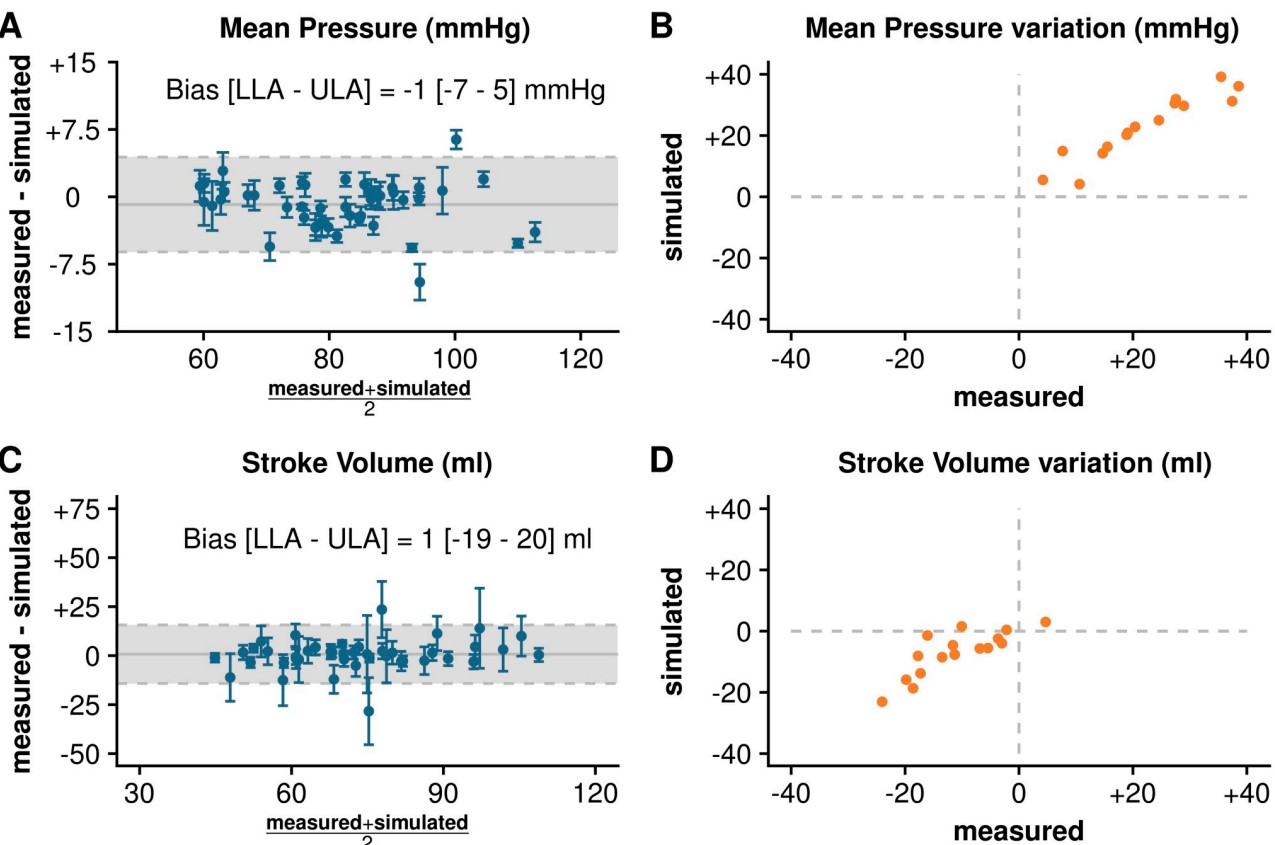

**Fig 4. Results of calibration.** Left: Bland-Altman plots for repeated measurements representing dispersion of the difference between measurement and simulation at baseline (n = 45 patients). Blue points and bars represent the mean and standard deviation for 10 heart beats in individual patients. Dashed horizontal lines represent the limit of agreement (±1.96 times standard deviation) and the horizontal gray line represents the bias or the mean difference between measurements and simulation. Right: 4-quadrant plots representing the variation of mean pressure and stroke volume from hypotension to maximum effect of norepinephrine in patients from hypotensive group (n = 16), orange points represent mean of ten beats for each patient. LLA, lower limit of agreement; ULA, upper limit of agreement.

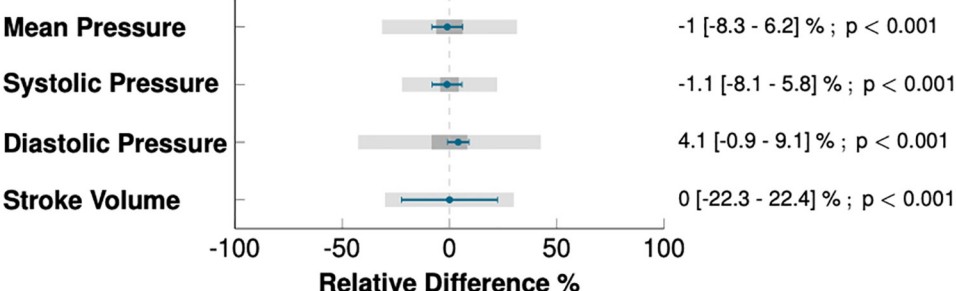

**Fig 5. Confidence intervals for the differences between measurements and simulations.** Dark gray boxes represent the equivalence area for the mean difference estimation (in percentage of the measured indicator), light gray boxes represent the equivalence area for confidence intervals. Limits of equivalence were defined as ±8mm Hg for pressure and ±30% for stroke volume, as recommended by international guidelines. Blue lines represent the mean and the confidence interval for the difference between measurement and simulation.

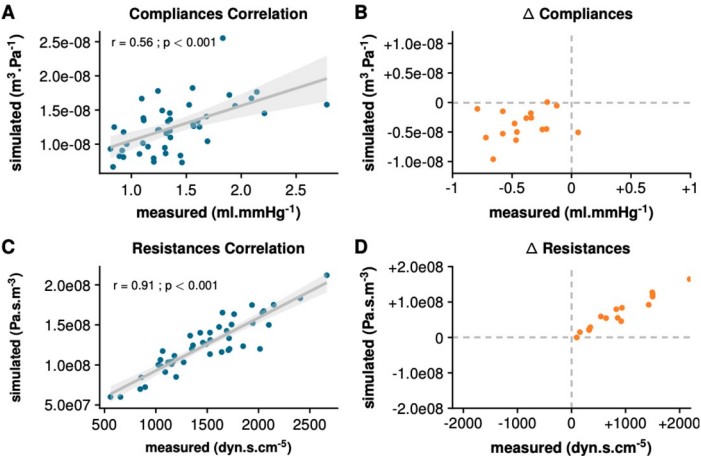

**Fig 6. Correlations between simulated and measured indicators.** A: Correlation plot representing simulated capacitance ($C_d$) against measured total arterial compliance ($C_{tot}$ = SV/PP) for all 45 patients at baseline. B: 4-quadrant plot of $\Delta C_d$ against $\Delta C_{tot}$ representing difference from hypotension to maximum effect of norepinephrine for 16 hypotensive patients. C: Correlation plot representing simulated resistance ($R_d$) and systemic vascular resistance ($SVR = \frac{MAP}{CO}$) for all 45 patients at baseline. D: 4-quadrant plot of $\Delta R_d$ against $\Delta SVR$ representing difference from hypotension to maximum effect of norepinephrine for 16 hypotensive patients. SV, stroke volume; PP, pulse pressure; MAP, mean aortic pressure.

exclusion zone for MAP and SV, respectively). The 4-quadrant plots in Fig 6 show the adequacy of the NOR-adjusted capacitance and resistance parameters with SVR and $C_{tot}$, respectively (>95% concordance with 10% exclusion zone, for both). To adjust the calibration, we had to significantly increase $R_d$ by 60±39%; $p < 0.001$, increase contractility by 14±11%; $p = 0.002$, and decrease $C_d$ by 27±16%; $p < 0.001$.

When analyzing the effect of blood pressure restoration by using the simulated PV loops (example in Fig 1), we observed an increase of $E_w$ by 13±12%; $p = 0.001$, and $I_w$ by 141±161%; $p < 0.001$, associated with a decrease of CE by 13±11%; $p < 0.001$. We also observed an increase of $V_{va}$ by 92±101% caused by an increase of arterial elastance ($E_a$) by 59±37%; $p < 0.001$, for both. Fig 7 shows the absolute variation of PV loop functional indicators between the baseline and the maximal effect of NOR.

## Validation group

The analysis of the validation group confronts the simulated vs measured PV loops and demonstrates the sensitivity to using the flow in ascending vs descending aorta to calibrate the Windkessel model. While Fig 8 shows a visual comparison, Table 6 demonstrates that the errors between the simulated and measured ventricular pressures and the quantities obtained from the PV loop analyses ($E_w$, CE and $V_{va}$ coupling) are all ≤10% in both types of recorded aortic flows.

## Discussion

This study demonstrates the feasibility of employing biomechanical modeling to augment CV physiological monitoring. The proposed framework allowed to set up models for 45 patients while using standard data recorded during neuroradiological procedures (without cardiac catheterization). The patient-specific models were subsequently used to quantify in vivo the CV consequences of IOH on the cardiovascular system (including PV loop analysis) as

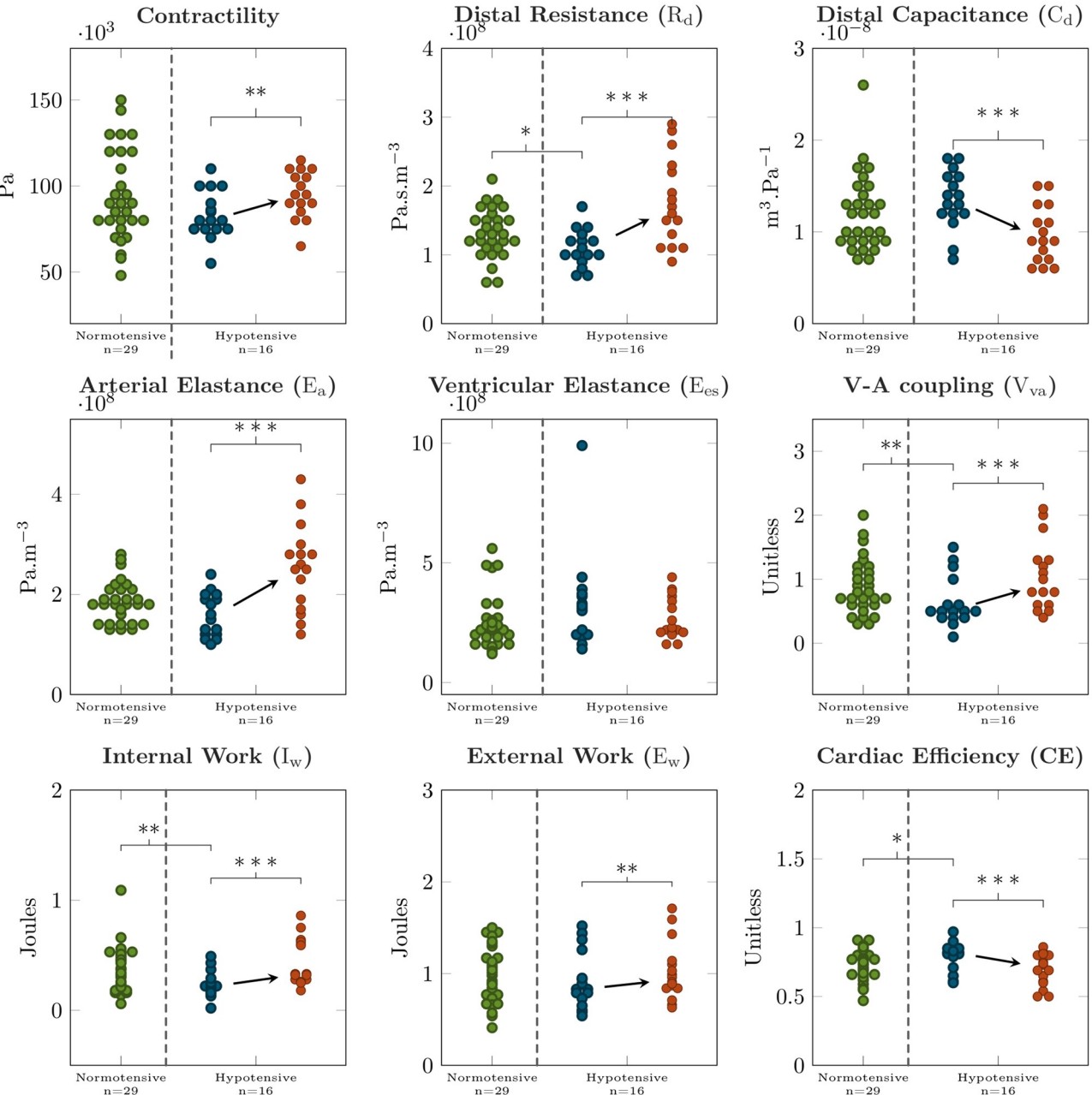

**Fig 7. Boxplots of model parameters and the results of the simulation.** Normotensive patients are in green; hypotensive patients before and after administering norepinephrine are in blue and red color, respectively. $^*$ $p < 0.05$; $^{**}$ $p < 0.01$; $^{***}$ $p < 0.001$.

compared to the normotensive population, and the effects of restoration of blood pressure by NOR administration.

First, we verified that the models were adequately calibrated in the cohort of 45 patients. We performed an equivalence study between the measurements and simulations by analyzing MAP, SAP, DAP and SV differences. The confidence interval s of the differences did not exceed the equivalence margins. Moreover, we observed that the numerical values prescribed for the parameters were in accordance with the expected theoretical levels. The validation

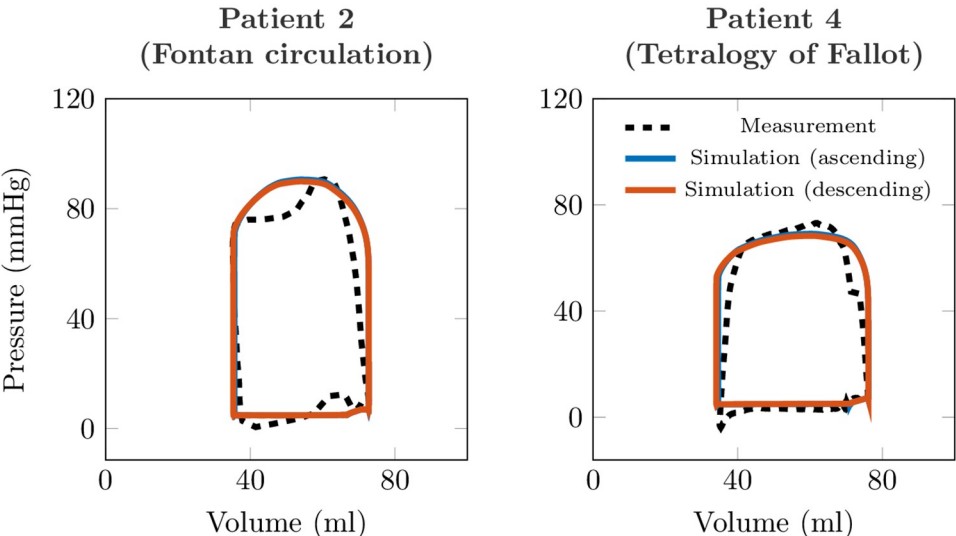

**Fig 8. Validation subjects.** Comparison of the simulated PV loops and of the measurements in selected validation subjects once descending aortic flow was used (as in our clinical study) or when directly measured ascending aortic flow was used.

group confirmed that using the proposed calibration procedure, the simulated and measured PV loops were in accordance.

We therefore assumed that the calibrated models behaved as the CV systems of the patients, which allows to access an advanced CV picture of the individual patients. Specifically, we were able to observe from a cardiac energetics viewpoint, that the hypotensive group expressed more efficient hearts with better ventricular-arterial coupling. Despite the higher efficiency, the model revealed an anesthetic drug-induced vasodilation. The restoration of blood pressure was required, as hypotension may worsen organ perfusion and lead to renal or myocardial ischemia [35, 36]. Optimizing the cardiac energetic expenditure could be the main target, however, in other situations (e.g. failing hearts or malignant hypertension management [7]).

Thirdly, we aimed to test the ability of our patient-specific model to quantify the changes in the CV system induced by a pharmacological challenge. NOR should enhance the myocardial contractility, increase the systemic vascular resistance and decrease the total arterial compliance [37, 38]. We can appreciate that, after NOR administration, both measured $C_{tot}$ and distal capacitance $C_d$ (model) varied in the expected direction, even though the correlation at

**Table 6. Relative error in simulation vs measurement (in %) in the validation group.** First and second line in each subject for measured flow in ascending, descending aorta, respectively. MP, mean ventricular pressure; SV, stroke volume; $E_w$, external work; A-V coupling, arterio-ventricular coupling; CE, cardiac efficiency.

| Patient | measurement type | MP | SV | $E_w$ | V-A coupling | CE |
|---|---|---|---|---|---|---|
| Patient 1 | ascending aorta | 10 | 6 | 2 | 6 | 4 |
| | descending aorta | 5 | 3 | 5 | 6 | 4 |
| Patient 2 | ascending aorta | 2 | 3 | 3 | 2 | 10 |
| | descending aorta | 2 | 2 | 3 | 5 | 10 |
| Patient 3 | ascending aorta | 1 | <1 | 4 | 2 | 1 |
| | descending aorta | 2 | 4 | 3 | 6 | 3 |
| Patient 4 | ascending aorta | 1 | 3 | 10 | 2 | 8 |
| | descending aorta | 1 | 1 | 10 | <1 | 8 |

baseline was mild. Measured SVR and distal resistance $R_d$ (model) were significantly correlated at baseline and at the maximal effect of NOR. We observed that the contractility increased and that cardiac energetic expenditure and the V-A coupling worsened. These observation s are compatible with the pharmacological effect s of NOR.

It has been shown that the model-estimated contractility correlates with the maximum upslope of the ventricular pressure $(dp/dt)_{max}$ [11, 16]. Moreover, a close relationship has been found between the contractility changes and the changes of maximum ventricular elastance $E_{es}$ in response to inotropic drugs [39]. While both $(dp/dt)_{max}$ and $E_{es}$ are only surrogate measures of contractility, which both in addition require ventricular catheterization, the estimated true myocardial contractility is likely to have a direct link to the energy needs of the cell.

Cardiovascular models combined with measured data have the potential to assist in diagnostic or therapeutic management by providing additional information not directly contained in the data. Patient-specific cardiovascular models are already available, see e.g. [40–45]. They have not yet been tested for monitoring of physiological functions, however. In [10], the authors used lumped-parameter modeling framework calibrated using 4D flow MRI to compute PV loops in 8 healthy volunteers. MRI is however not suitable for monitoring. The authors of [46] explored the effect of sodium nitroprusside on PV loops using a lumped-parameter model in 5 patients with decompensated heart failure. Their calibration involved a pulmonary artery catheter, which however narrows the applicability of the method.

## Limitations

Further steps will need to be performed prior to employing the model-augmented monitoring in routine practice. First, the combination of our biophysical modeling framework with models based on time-varying elastance as in [47] is currently being investigated (as suggested in [48]) to achieve real-time monitoring, compared with simulation runs of about ten seconds (on a standard laptop) in the present work. Secondly, not having access to the ventricular or atrial pressure, we used a validated semi-quantitative method to classify LV filling pressure as a high or normal preload [30]. Moreover, the filling pressure was kept at the same level during NOR administration, even though NOR is known to increase the LV filling pressure [49]. This approximation is likely to have led to an overestimation of contractility in response to NOR. In future work s we will investigate a possibility of including a venous return model, while keeping the protocol still clinically acceptable. The flow in the ascending aorta was replaced by the descending aorta flow waveform scaled by using the measured SV. This could be considered as a surrogate measure for the ascending aorta flow if no significant aortopathy is present (as was the case in our cohort), which was confirmed by using our validation subjects. A significant stenosis in the aortic arch (such as in aortic coarctation) would require to include information about the stenosis level (obtained e.g. by TTE prior to the procedure) into consideration. Finally, the setting of our proof-of-concept study involves aortic pressure measurement. Although it can be easily obtained during radiological interventions, it is not routinely available during GA as only a peripheral artery cannula is typically requested for medical concerns. A transfer function between peripheral arterial pressures and aortic pressures will be included in our future work [50, 51].

## Conclusions

This study aimed at evaluating the feasibility of hemodynamic monitoring augmented by employing a patient-specific biomechanical model of heart and circulation set up using routine hemodynamic measurements during neuroradiological procedure. Our framework allowed to create biomechanical models specific for individual patients. Such models then allowed the

interpretation of patient data comparatively between normotensive and hypotensive state by plotting simulated PV loops and by quantitative estimation of pharmacologically-induced alterations of the cardiovascular system. Even though further methodological improvements are needed to transfer the technology to the bedside, the presented work represents a significant step towards augmenting cardiovascular monitoring by using biophysical modeling. The availability of such ready-to-use numerical patient-specific models has the potential to cause a paradigm shift in physiological monitoring and management of patients in critical states during complex general anesthesia procedures or at intensive care units throughout the medical specializations.

## Supporting information

**S1 File. Data file *S1 File.txt* contains data in the form of time-vs-aortic pressure and flow (for all subjects in the study), as well as in addition the ventricular volumes and pressures for the validation subjects. For each subject, the left ventricular end-diastolic and end-systolic volumes, and myocardial mass are also indicated.**
(ZIP)

## Acknowledgments

We would like to thank Dr Philippe Moireau (Inria Saclay Ile-de-France) for his support concerning the model implementation and Mr José Serrano (Lariboisière hospital, Paris, France) for his involvement in collecting the data.

## Author Contributions

**Conceptualization:** Arthur Le Gall, Fabrice Vallée, Dominique Chapelle, Étienne Gayat, Radomír Chabiniok.

**Data curation:** Arthur Le Gall, Fabrice Vallée, Kuberan Pushparajah, Tarique Hussain, Étienne Gayat.

**Formal analysis:** Arthur Le Gall.

**Funding acquisition:** Fabrice Vallée, Kuberan Pushparajah, Tarique Hussain, Alexandre Mebazaa, Dominique Chapelle, Étienne Gayat.

**Methodology:** Arthur Le Gall, Fabrice Vallée, Alexandre Mebazaa, Dominique Chapelle, Étienne Gayat, Radomír Chabiniok.

**Project administration:** Arthur Le Gall, Fabrice Vallée, Dominique Chapelle, Étienne Gayat, Radomír Chabiniok.

**Resources:** Alexandre Mebazaa, Dominique Chapelle, Étienne Gayat.

**Software:** Arthur Le Gall, Dominique Chapelle, Radomír Chabiniok.

**Supervision:** Fabrice Vallée, Dominique Chapelle, Étienne Gayat, Radomír Chabiniok.

**Validation:** Kuberan Pushparajah, Tarique Hussain, Alexandre Mebazaa, Dominique Chapelle.

**Visualization:** Arthur Le Gall.

**Writing – original draft:** Arthur Le Gall, Radomír Chabiniok.

**Writing – review & editing:** Arthur Le Gall, Fabrice Vallée, Kuberan Pushparajah, Tarique Hussain, Alexandre Mebazaa, Dominique Chapelle, Étienne Gayat, Radomír Chabiniok.

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
