## [Decision Letter · Decision Letter 0]

2 Apr 2020

PONE-D-20-06385

Monitoring of cardiovascular physiology augmented by a patient-specific biomechanical model during general anesthesia. A proof of concept study.

PLOS ONE

Dear Dr. Radomír Chabiniok

Thank you for submitting your manuscript to PLOS ONE. After careful consideration, we feel that it has merit but does not fully meet PLOS ONE’s publication criteria as it currently stands. Therefore, we invite you to submit a revised version of the manuscript that addresses the points raised during the review process.

I would appreciate if you response to the reviewers' comments 

We would appreciate receiving your revised manuscript by May 17 2020 11:59PM. To enhance the reproducibility of your results, we recommend that if applicable you deposit your laboratory protocols in protocols.io, where a protocol can be assigned its own identifier (DOI) such that it can be cited independently in the future. For instructions see: http://journals.plos.org/plosone/s/submission-guidelines#loc-laboratory-protocols

We look forward to receiving your revised manuscript.

Kind regards,

Ehab Farag, MD FRCA FASA

Academic Editor

PLOS ONE

2. Please note that PLOS ONE has specific guidelines on software sharing (https://journals.plos.org/plosone/s/materials-and-software-sharing#loc-sharing-software) for manuscripts whose main purpose is the description of a new software or software package. In this case, new software must conform to the Open Source Definition (https://opensource.org/docs/osd) and be deposited in an open software archive. Please see https://journals.plos.org/plosone/s/materials-and-software-sharing#loc-depositing-software for more information on depositing your CardiacLab library, if you have not already done so.

4. We note that you have a patent relating to material pertinent to this article. Please provide an amended statement of Competing Interests to declare this patent (with details including name and number), along with any other relevant declarations relating to employment, consultancy, patents, products in development or modified products etc. Please confirm that this does not alter your adherence to all PLOS ONE policies on sharing data and materials, as detailed online in our guide for authors http://journals.plos.org/plosone/s/competing-interests by including the following statement: "This does not alter our adherence to  PLOS ONE policies on sharing data and materials.” If there are restrictions on sharing of data and/or materials, please state these. Please note that we cannot proceed with consideration of your article until this information has been declared.

Reviewers' comments:

Reviewer's Responses to Questions

**Comments to the Author**

1. Is the manuscript technically sound, and do the data support the conclusions?

Reviewer #1: Yes

Reviewer #2: Yes

2. Has the statistical analysis been performed appropriately and rigorously? 

Reviewer #1: Yes

Reviewer #2: I Don't Know

3. Have the authors made all data underlying the findings in their manuscript fully available?

Reviewer #1: Yes

Reviewer #2: Yes

4. Is the manuscript presented in an intelligible fashion and written in standard English?

Reviewer #1: Yes

Reviewer #2: Yes

5. Review Comments to the Author

Reviewer #1: Very well done and interesting study.As the authors describe,paves the way for future studies based on their modeling.Good statistical analysis.Diagrams are well done too.No further changes recommended.

Reviewer #2: This interesting and innovative study sought to improve the state-of-the-art of hemodynamic monitoring by developing a method for constructing patient-specific bio-mechanical models of the heart and circulation. Such models may offer future clinicians a step forward in the physiological monitoring and pharmacological management of ICU patients and patients undergoing general anesthesia.

The science behind the author’s efforts is quite complex, but adequately described.

However, a major concern I have is the confusing use of English language throughout the manuscript. Two examples are the following:

around 230,000,000 general anesthesia (GA) are performed each year worldwide

and

titration of saline isochloride by 250 ml (Do you mean Normal Saline?)

The authors note that the study “simulations were performed using the CardiacLab library”, an in-house developed MATLAB model. Is this MATLAB code available for download by other investigators?

One final suggestion I have is to make more explicit the limitations of the present study in a separate section.

6. PLOS authors have the option to publish the peer review history of their article (what does this mean?). If published, this will include your full peer review and any attached files.

Reviewer #1: No

Reviewer #2: No

---

## [Author Response · Author response to Decision Letter 0]

21 Apr 2020

Answer: We revised journal requirements and adapted the style and file naming accordingly. 

2. Please note that PLOS ONE has specific guidelines on software sharing (https://journals.plos.org/plosone/s/materials-and-software-sharing#loc-sharing-software) for manuscripts whose main purpose is the description of a new software or software package. In this case, new software must conform to the Open Source Definition (https://opensource.org/docs/osd) and be deposited in an open software archive. Please see https://journals.plos.org/plosone/s/materials-and-software-sharing#loc-depositing-softwarefor more information on depositing your CardiacLab library, if you have not already done so.

Answer:

The purpose of the article is not to describe a new software or software package. The model used in this paper is fully described within the manuscript and the cited material (in particular [Caruel et al, BMMB2014], [Sainte-Marie_C&S2006], and [Chapelle et al, INT J MULTISCALE COM2012]). CardiacLab is just an internal code and we therefore retract this internal name from the manuscript. The modified closing paragraph of the Methods therefore reads:

"The simulations were performed using an in-house implementation of the model in MATLAB (The MathWorks Inc, Natick, Massachusetts).”

All data are available within the Supplementary material S1_file.txt. The Data Availability statement is therefore reads:

"All relevant data are within the manuscript and its Supporting Information files."

4. We note that you have a patent relating to material pertinent to this article. Please provide an amended statement of Competing Interests to declare this patent (with details including name and number), along with any other relevant declarations relating to employment, consultancy, patents, products in development or modified products etc. Please confirm that this does not alter your adherence to all PLOS ONE policies on sharing data and materials, as detailed online in our guide for authors http://journals.plos.org/plosone/s/competing-interestsby including the following statement: "This does not alter our adherence to PLOS ONE policies on sharing data and materials.” If there are restrictions on sharing of data and/or materials, please state these. Please note that we cannot proceed with consideration of your article until this information has been declared.

Answer:

We would like to ammend our competing interest statement: 

“A.L.G., F.V, D.C. and R.C. are co-owners of the patent entitled "Dispositif cardiaque" (number 1758006, 2017). A research license agreement is currently ongoing between the Anesthesiology and intensive care department of Lariboisi\\`ere hospital, Paris, France and Deltex Medical, Chichester, UK. “

to: 

“I have read the journal's policy and the authors of this manuscript have the following competing interests: A.L.G., F.V, D.C. and R.C. are co-owners of the patent entitled "Dispositif cardiaque" (number 1758006, 2017). A research license agreement is currently ongoing between the Anesthesiology and intensive care department of Lariboisi\\`ere hospital, Paris, France and Deltex Medical, Chichester, UK. This does not alter our adherence to PLOS ONE policies on sharing data and materials.”

Comments of reviewers:

Reviewer #1: Very well done and interesting study.As the authors describe,paves the way for future studies based on their modeling.Good statistical analysis.Diagrams are well done too.No further changes recommended.

Answer: We are grateful for such a positive feedback. 

Reviewer #2: This interesting and innovative study sought to improve the state-of-the-art of hemodynamic monitoring by developing a method for constructing patient-specific bio-mechanical models of the heart and circulation. Such models may offer future clinicians a step forward in the physiological monitoring and pharmacological management of ICU patients and patients undergoing general anesthesia.

The science behind the author’s efforts is quite complex, but adequately described.

However, a major concern I have is the confusing use of English language throughout the manuscript. Two examples are the following:

around 230,000,000 general anesthesia (GA) are performed each year worldwide

and

titration of saline isochloride by 250 ml (Do you mean Normal Saline?)

Answer: 

The manuscript was proofread in detail and English ameliorated (the changes are indicated in the manuscript with track changes). In particular: 

“around 230,000,000 general anesthesia (GA) are performed each year ...” was replaced by 

“ around 230 million major surgical procedures under general anesthesia (GA) are performed each year ….”

And

" titration of saline isochloride by 250 ml step” was replaced by 

"titration of saline solution by 250 ml steps " 

Reviewer #2: The authors note that the study “simulations were performed using the CardiacLab library”, an in-house developed MATLAB model. Is this MATLAB code available for download by other investigators?

Answer: The purpose of the article is not to describe a new software or software package. The model used in this paper is fully described within the manuscript and the cited material (in particular [Caruel et al, BMMB2014], [Sainte-Marie_C&S2006], and [Chapelle et al, INT J MULTISCALE COM2012]). CardiacLab is just an internal code and we therefore retract this internal name from the manuscript. The modified closing paragraph of the Methods therefore reads:

"The simulations were performed using an in-house implementation of the model in MATLAB (The MathWorks Inc, Natick, Massachusetts).”

Reviewer #2: One final suggestion I have is to make more explicit the limitations of the present study in a separate section."

Answer: In the revised manuscript, we extended the limitations and presented them in a separate section.

---

## [Editor Report · Decision Letter 1]

23 Apr 2020

Monitoring of cardiovascular physiology augmented by a patient-specific biomechanical model during general anesthesia. A proof of concept study.

PONE-D-20-06385R1

Dear Dr.

   Radomír Chabiniok 

We are pleased to inform you that your manuscript has been judged scientifically suitable for publication and will be formally accepted for publication once it complies with all outstanding technical requirements.

With kind regards,

Ehab Farag, MD FRCA FASA

Academic Editor

PLOS ONE
---

## [Editor Report · Acceptance letter]

1 May 2020

PONE-D-20-06385R1 

Monitoring of cardiovascular physiology augmented by a patient-specific biomechanical model during general anesthesia. A proof of concept study. 

Dear Dr. Chabiniok:

I am pleased to inform you that your manuscript has been deemed suitable for publication in PLOS ONE. Congratulations! Your manuscript is now with our production department. 

With kind regards,

on behalf of

Dr. Ehab Farag 

Academic Editor

PLOS ONE